# Calcium Role in Gap Junction Channel Gating: Direct Electrostatic or Calmodulin-Mediated?

**DOI:** 10.3390/ijms25189789

**Published:** 2024-09-10

**Authors:** Camillo Peracchia

**Affiliations:** Department of Pharmacology and Physiology, School of Medicine and Dentistry, University of Rochester, Rochester, NY 14642-8711, USA; camillo.peracchia@gmail.com or camillo_peracchia@urmc.rochester.edu

**Keywords:** calcium, calmodulin, gap junctions, connexin, innexin, channel gating, gating, cell–cell coupling, cell communication

## Abstract

The chemical gating of gap junction channels is mediated by cytosolic calcium (Ca^2+^_i_) at concentrations ([Ca^2+^]_i_) ranging from high nanomolar (nM) to low micromolar (µM) range. Since the proteins of gap junctions, connexins/innexins, lack high-affinity Ca^2+^-binding sites, most likely gating is mediated by a Ca^2+^-binding protein, calmodulin (CaM) being the best candidate. Indeed, the role of Ca^2+^-CaM in gating is well supported by studies that have tested CaM blockers, CaM expression inhibition, testing of CaM mutants, co-localization of CaM and connexins, existence of CaM-binding sites in connexins/innexins, and expression of connexins (Cx) mutants, among others. Based on these data, since 2000, we have published a Ca^2+^-CaM-cork gating model. Despite convincing evidence for the Ca^2+^-CaM role in gating, a recent study has proposed an alternative gating model that would involve a direct electrostatic Ca^2+^-connexin interaction. However, this study, which tested the effect of unphysiologically high [Ca^2+^]_i_ on the structure of isolated junctions, reported that neither changes in the channel’s pore diameter nor connexin conformational changes are present, in spite of exposure of isolated gap junctions to [Ca^2+^]_i_ as high at the 20 mM. In conclusion, data generated in the past four decades by multiple experimental approaches have clearly demonstrated the direct role of Ca^2+^-CaM in gap junction channel gating.

## 1. Introduction

Data reporting that direct cell communication can be reduced to total electrical and metabolic cell uncoupling were first reported almost one hundred years before direct cell communication via gap junction channels was discovered. Indeed, Engelmann reported that healthy cardiac cells cease to communicate with neighboring damaged cells [1]. This phenomenon, named “healing-over”, must have surprised him because the cardiac muscle was thought to be a syncytium [2]; it took three quarters of a century for the healing-over phenomenon to finally be confirmed [3,4,5,6]. This mechanism, now known as “cell–cell uncoupling”, is based on chemical gating of gap junction channels; rev. in [6,7,8,9]. Channel gating is activated by changes in the cytosolic [Ca^2+^]_i_ or [H^2+^]_i_, as well as by trans-junctional voltage (Vj) gradients. While a model envisioning a direct Ca^2+^ effect on channel gates has been published [10,11], data from many studies, including our own, have strongly supported the intermediary role of Ca^2+^-activated calmodulin (CaM); rev. in [12,13].

## 2. Cytosolic Calcium (Ca^2+^_i_) and Gap Junction Channel Regulation

Jean Délèze published the earliest data supporting the Ca^2+^_i_ role in channel gating by reporting that cut cardiac fibers do not uncouple from neighboring damaged fibers in the absence of extracellular calcium (Ca^2+^_o_) [14]. Soon after, the Ca^2+^ role in channel gating was supported by evidence that direct cell–cell communication ceases with a rise in cytosolic [Ca^2+^]_i_ [7,8,15,16,17,18]. Rose and Loewenstein were first to provide clear evidence for the role of Ca^2+^_i_ in gap junction channel gating by showing that uncoupling parallels a [Ca^2+^]_i_ rise in the cytosol adjacent to gap junctions, monitored by the Ca^2+^-sensitive probe aequorin [16,17]. The numerous studies that have followed have further proven the Ca^2+^_i_ role in gap junction channel gating: rev. in [9,18,19].

### [Ca^2+^]_i_ Affecting Gating of Gap Junction Channel

A few studies suggested that [Ca^2+^]_i_ as great as 40–400 µM are needed for uncoupling either wounded insect gland cells [15], intracellularly perfused Fundulus blastomeres [20] or neonatal cardiac cells [21]. In contrast, more numerous studies have demonstrated the effectiveness of much lower [Ca^2+^]_i_, ranging from high nanomolar (nM) to low micromolar (µM). These low [Ca^2+^]_i_ were first reported to be effective in Chironomus salivary gland cells [17,18,22] and mammalian cardiac fibers [23]. Similarly, Dahl and Isenberg reported that channel gating is induced in sheep cardiac fibers by [Ca^2+^]_i_ > 500 nM [24].

Two famous studies have demonstrated that [Ca^2+^]_i_ as low as 251 nM are effective in cardiac cell pairs, in which one cell was ruptured to enable the influx of bathing solutions buffered for Ca^2+^ and pH [25,26]. Recently, Dekker and coworkers [27] reported that in perfused rabbit papillary muscles the application of ionomycin and gramicidin causes uncoupling at [Ca^2+^]_i_ = 685 nM; similar [Ca^2+^]_i_ were effective in uncoupling caused by ischemia/reperfusion. High nM to low μM [Ca^2+^]_i_ also activated channel gating in crayfish lateral giant axons [28], rat lacrimal cells [29], cells of Novikoff hepatoma [30,31], astrocytes [32,33,34], cultured cells of eye lens [35], pancreatic β-cells [36], human fibroblasts [37], and cultured cells expressing connexin43 (Cx43) [38].

In pancreatic and lacrimal cells, uncoupling developed as secretion was induced by application of acetylcholine or different secretagogues, tested at lower concentrations than those causing total enzyme secretion [39,40,41], and by membrane depolarization or accumulation of cyclic nucleotides [42,43]. Indeed, in pancreatic acinar cells, the application of secretagogues at concentrations sufficient to cause maximal secretion only increased [Ca^2+^]_i_ from 180 nM to 860 nM [44].

In astrocytes co-injected with Ca^2+^ and Lucifer Yellow CH, a water-soluble fluorescent dye, nM [Ca^2+^]_i_ activated gating and prevented cell–cell dye transfer independently of pHi [32]. Blockage of dye transfer was linearly correlated to [Ca^2+^]_i_ ranging between 150 nM and 600 nM [32]. In agreement with these findings are data showing that the addition of 20mM of 1,2-bis(o-aminophenoxy)ethane-N,N,N′,N′-tetraacetic acid (BAPTA) to the solution of patch pipettes greatly enhances the astrocytes’ coupling [33], suggesting that gating is even sensitive to basal [Ca^2+^]_i_; indeed, there is evidence that several gap junction channels are in closed state even at resting [Ca^2+^]_i_ (see in the following). Dye coupling ceased in ionomycin-treated astrocytes when [Ca^2+^]_i_ was increased to 500 nM [34], and comparable data were published for cultured eye lens cells [45]. In Neuro-2a cells (N2a) expressing Cx43, ionomycin addition to superfusion solutions raised Ca^2+^ influx and inhibited junctional conductance (Gj) by 95% [38]. Ionomycin caused [Ca^2+^]_i_ to rise from ~80 nM to ~250 nM, further confirming that Ca^2+^i is a fine regulator of cell coupling [38].

[Ca^2+^]_i_ as low as 0.5–1 µM prevented cell-to-cell spread of the dye Lucifer Yellow CH in cell cultures of embryonic chicken [35]; exposure of these cells to A23187, a Ca-ionophore, or ionomycin, increased [Ca^2+^]_i_ from ~110 to ~400 nM, and the cell–cell diffusion of the dye was dramatically curtailed. Subsequent superfusion with no-Ca-added solutions containing EGTA decreased [Ca^2+^]_i_ to ~260 nM and dye transfer recovered [35]. Nanomolar [Ca^2+^]_i_ also dramatically decreased Gj in pancreatic β-cells, in which a rise in [Ca^2+^]_i_ was caused by dropping the temperature from 37o to 30o and by raising [Ca^2+^]o from 2.56 to 7.56 mM [36].

To assess the [Ca^2+^]_i_ that activates gating in detail, we tested Cx43-expressing Novikoff hepatoma cells by dual-whole-cell clamp electrophysiology (Figure 1) [30,31]. In Novikoff cells, Ca^2+^ gating sensitivity was evaluated by monitoring the Gj drop at different BAPTA-buffered [Ca^2+^]_i_, at pHi = 7.2 or 6.1, buffered by 4-(2-hydroxyethyl)-1-piperazineethanesulfonic acid (HEPES) and 2-[N-morpholino]ethanesulfonic acid (MES), respectively. In these cells, gating was affected by [Ca^2+^]_i_ ranging from 500 nM to 1 µM at either pHi (Figure 1A). With [Ca^2+^]_i_ = 120 nM or lower, Gj dropped from 100% to 40–50% with mean τ’s = 35.2 and 22.3 min, at pHi = 6.1 or 7.2, respectively; this is the usual Gj drop in cells studied by double-whole-cell clamp. With [Ca^2+^]_i_ ranging from 500 nM to 1.0 µM, Gj decreased to ~25% of resting values with mean τ’s = 5.9 min and 6.2 min, at pHi = 6.1 and 7.2, respectively (Figure 1A). With [Ca^2+^]_i_ = 3 µM (pH = 7.2) the cells became uncoupled in less than 1 min (τ = ~20 s; Figure 1B). These data confirmed earlier evidence that the gap junction channel gates of Cx43 are sensitive to [Ca^2+^]_i_ in the high nanomolar to low µM range; rev. in [13,19].

High nanomolar [Ca^2+^]_i_ was also effective in gating Novikoff hepatoma cell pairs briefly treated with 20 µM arachidonic acid (AA) [1]. AA is a fatty acid generated by phospholipase A2 [46], with receptor-mediated or independent hypoxia-induced stimulation. AA is metabolized to prostaglandins and thromboxanes by cyclo-oxygenases, and to leukotrienes and hydroxyeicosatetraenoic acid by lipoxygenases. These compounds, eicosanoids, modulate signal transduction in several cell systems [47,48,49]. Thromboxane A2, a cyclooxygenase breakdown product of AA, is a relevant factor in AA-induced [Ca^2+^]_i_ rise.

Twenty-second applications of AA uncoupled the cells rapidly and reversibly (Figure 2 and Figure 3). Uncoupling was prevented by the addition of BAPTA to the patch pipette solution (Figure 2). In contrast, ethylene glycol-bis (β-aminoethyl ether)-N,N,N′,N′-tetraacetic acid (EGTA), a less efficient cytosolic Ca^2+^ buffer, was significantly less effective (Figure 3). Similarly, 20 s exposures to AA did not affect coupling in cells superfused with no-Ca^2+^-added solutions, proving that uncoupling results from Ca^2+^ influx [1]. Parallel experiments in which [Ca^2+^]_i_ was measured with Fura-2 demonstrated that in cells superfused with normal [Ca^2+^]o solutions, [Ca^2+^]_i_ rises to 0.7–1.5 µM with 20 s AA treatments. The [Ca^2+^]_i_ rise was caused by AA-induced Ca^2+^ influx, because [Ca^2+^]_i_ increased minimally in cells superfused with Ca^2+^-free solutions without affecting Gj [1]. Significantly, however, prolonged AA treatment (1 min, 20 µM) slowly uncoupled the cells even in Ca^2+^-free media, suggesting that AA has a rapid Ca^2+^-dependent effect as well as a slow Ca^2+^-independent effect on gating [1], which is like that caused by exposure to anesthetics.

In conclusion, over the past half-century, the effectiveness on gap junction channel gating of [Ca^2+^]_i_ ranging from high nM to low μM has been reported by multiple scientists in dozens of studies performed on a variety of cell systems of vertebrates and invertebrates. Significantly, a similar cytosolic [Ca^2+^]_i_ range has been reported to gate connexin hemichannels as well [50]. Since gap junction proteins such as connexins in vertebrates and innexins in invertebrates do not possess high-affinity Ca^2+^-binding sites, it is obvious that a Ca^2+^-modulated protein containing typical EF-hand Ca^2+^-binding sites [51,52,53] is needed to mediate [Ca^2+^]_i_-induced gating. Indeed, since the early 1980s, many studies have supported a direct role of Ca^2+^-activated calmodulin in mediating Ca^2+^-induced gating of gap junction channels; rev. in [12,13] (see in the following).

## 3. Does Calcium Act Directly on Gap Junction Channel Gating?

A direct Ca^2+^ role in on gap junction channel gating at [Ca^2+^]_i_ in the high nM to low μM range (see in the previous chapter) requires highly sensitive Ca^2+^-binding sites such as those present in Ca^2+^-modulated proteins (EF-hand domains) [51,52,53], none of which are expressed in connexins/innexins. Despite this, a recent study [10,11] has proposed a direct [Ca^2+^]_i_ role in chemical gating of Cx26 channels that would involve “An electrostatic mechanism for Ca^2+^-mediated regulation of gap junction channels” [11]. This mechanism envisions that gating results from Ca^2+^-mediated interlinking of neighboring connexin monomers at three Ca^2+^-coordination sites located at the first extracellular loop (E1). These sites comprehend two residues of one monomer (G45 and E47) and one residue (E42) of the neighboring monomer (Figure 4).

Based on a remarkable computational analysis [11], the authors concluded that the Ca^2+^-interlinking of these residues forms a positively charged electrostatic barrier sufficient to prevent channel diffusion of K+ ions. However, it is worth noting that Ca^2+^-binding was not found to close the channel’s pore and did not induce significant conformational changes in the channel structure; in the authors words: “Ca^2+^ binding to the Cx26 GJC (Gap Junction Channel) did not trigger a global conformational change to a sterically closed state, as observed in low-resolution studies of Cx GJCs3 and hemichannels34. As Cx26 was co-crystallized with Ca^2+^, we assumed that large-scale structural rearrangements were possible and would have occurred before crystallization. However, we observed only local conformational changes in the vicinity of Ca^2+^ coordination, which did not occlude the pore” [11].

While it is very likely that at [Ca^2+^]_i_ as huge as 20 mM, as tested in this study [11], Ca^2+^ interlinks these sites, the extrapolation of this interaction to the in vivo gating of cell–cell channels, which occurs at high nM to low µM [Ca^2+^]_i_ (see in a previous chapter) is a huge flight of imagination for many reasons. First of all, the no-Ca-added saline used in control conditions [11], which did not cause Ca^2+^ binding to the connexon sites, contained 20 mM sodium formate, whose Ca^2+^ stability constant is ~0.8 [54], and did not include Ca^2+^ chelators such as ethylenediaminetetraacetic acid (EDTA), EGTA, or BAPTA. Therefore, the [Ca^2+^] of the saline used as control must have been far greater than 12 µM. So, in the no-Ca-added saline, the [Ca^2+^] was undoubtedly much higher than the [Ca^2+^]_i_ known to activate gap junction channel gating (see in a previous chapter).

It is also worth noting that the Cx26’s residue E42, which corresponds to E44-MA in connexins’ multiple alignment (MA; Figure 4) is present in only 4 out of 22 connexins (Figure 4); even if one adds 3 connexins with another acidic residue (D44-MA; Figure 4), only~32% of connexins contain acidic residues in this location. Indeed, the authors noted the lack of E44-MA in most connexins, yet they still supported the validity of their model by mentioning that other connexins have an E41 residue (E43-MA). However, they failed to mention that a shift from the E42 (E44-MA) to the E41 (E43-MA) position, without a similar shift of the most relevant G45 (G47-MA) and E47 (E49-MA), could hinder the Ca^2+^ ability to cross-link them to E41 of the adjacent connexin monomer, as the shift of one position in the alpha-helix would cause an ~100o angle shift. In addition, human Cx30.2 does not contain acidic residues in this or nearby locations (Figure 4). Furthermore, the relevant G45 residue (G47-MA) of Cx26 is absent in four connexins: Cx25h, Cx32r, Cx45m, and Cx50h, while it is replaced by an acidic residue in six connexins (Figure 4).

Even though human Cx40 and Cx43 are virtually identical in the “ExxxGDE” sequence (res. 42–48; MA res. 43–49), Xu and coworkers [38] have reported that only channels made of Cx43 are Ca^2+^-gated (via CaM activation); rev. in [12]. Based on the proposed model [10,11] this would be paradoxical, because Cx43 and human Cx40 channels should display the same Ca^2+^-gating sensitivity.

The authors reported that despite Ca^2+^-binding to the site, the diameter of the pore in unaltered; in their words: “we did not observe physical closure of the channel upon Ca^2+^ binding” [11]; therefore, they concluded that only cations are blocked by “substantial positive electrostatic surface potential in the Ca^2+^-bound structure“ [11], while cell–cell diffusion of anions and neutral molecules should be unrestricted. Obviously, this contradicts data from many studies that have proven the [Ca^2+^]_i_-induced block of cell–cell diffusion of acidic fluorescent dyes such as fluorescein (MW = 330) [18,22,55,56]. More importantly, if this Ca^2+^-gating mechanism were only blocking cell–cell diffusion of cations, with cell damage, all neutral and negatively charged molecules would rapidly diffuse from healthy cells to the damaged cell and to the extracellular medium. This contradicts the basic function of Ca^2+^-induced channel gating—the Ca^2+^-dependent safety mechanism named “healing over” [1,3,4,5,6,14,57,58,59]. Moreover, connexins like Cx43, for example, are non-selective [60]; therefore, this [Ca^2+^]_i_ gating mechanism [10,11] would not block Cl- diffusion, and consequently would not cause electrical uncoupling. Indeed, this is disproven by numerous studies that have demonstrated electrical uncoupling (total channel closure) by activation of Cx43’s chemical gate [38,61]. In addition, cells coupled by Cx30.2, which lack acidic residues at E42 or E41 (MA E44 or E43, respectively; Figure 4), would be totally unprotected by the healing-over mechanism—cell–cell uncoupling would not take place.

Significantly, evidence from this study proving that the pore diameter is unaltered and that large conformational changes in the channel are absent with application of [Ca^2+^]_i_ as high at the 20 mM [10,11] indirectly confirms the idea that a Ca^2+^-modulated protein, most likely CaM, is required for mediating Ca^2+^-induced gap junction channel gating. One should also keep in mind that despite marvelous crystallographic works as this [10,11], making predictions of in vivo gating mechanisms is unwise because isolated junctions are mere “skeletons” of gap junctions, as they obviously must have lost most or all of the accessory molecules, including CaM, during the isolation procedure.

## 4. Calmodulin Role in Channel Gating

Numerous studies have confirmed evidence that calmodulin (CaM) is the soluble intermediate of Ca^2+^-induced channel gating; rev. in [12,19]. Data supporting the role of a soluble intermediate were first reported in 1981 by Johnston and Ramón who demonstrated that lateral giant axons of crayfish do not uncouple when they are intracellularly perfused with solutions high in [Ca^2+^] or [H^+^] [62]. In the early 1980s, we proposed CaM as the intermediate of Ca^2+^-induced cell–cell channel gating [63]. Consistent with our CaM hypothesis was evidence that CaM binds to the gap junction protein of the liver (Cx32) and to gap junctions of crayfish’s hepatopancreas [64,65].

Data showing that CaM is indeed the molecular intermediate that was washed away by the intracellularly perfused axons were provided by Arellano and coworkers [66], who reported that in crayfish axons perfused intracellularly with Ca^2+^-CaM at pCa 5.5, the junctional resistance (Rj) increases from ~60 kΩ to 500–600 kΩ, while in axons perfused with CaM in low Ca^2+^-saline (pCa > 7), with CaM-free high Ca^2+^ saline (pCa 5.5) or with Ca-free saline, Rj remained at resting values [66]. Innexins rather than connexins are expressed in crayfish, but innexins are like connexins and contain CaM-binding sites. In crayfish, both innexin-1 and innexin-2 are expressed [67]. Innexins contain CaM-binding domains at both CT and CL2 (Figure 5). The role of a soluble intermediate in channel gating was also confirmed by evidence that intracellularly perfused *Xenopus* oocytes expressing Cx43 are insensitive to acidification [68], as well as by the fact that neither high [H^+^] nor high [Ca^2+^] solutions close Cx32 hemichannels inserted into liposomes [69].

In the past four decades, the CaM role in cell–cell gating has been strongly supported by evidence generated by several approaches, such as exposure to CaM inhibitors [38,45,63,70,71,72,73,74], CaM expression inhibition [75,76,77], expression of a Ca^2+^-hypersensitive CaM mutant [78,79], colocalization of CaM and connexins [78,79,80], and evidence for CaM binding to connexins [64,65,78,79] and to connexin peptides matching the CaM-binding sites of several connexins [12,38,80,81,82,83,84,85,86]; rev. in [12,19].

CaM is composed of 148 amino acids and is a very well-preserved protein expressed in plants and animals. It is a 65 Å long protein and contains two spherical lobes of ~35 × 25 Å in diameter, known as the N- and C-lobe [87]. The NH2 terminus is linked to the N-lobe, which is connected to the C-lobe by a flexible tether. Each lobe contains two specialized sequences, known as EF-hands [52], with nM affinity to Ca^2+^. The C-lobe has Ca^2+^ affinity approximately one order of magnitude greater than that of the N-lobe. Ca^2+^ interaction with Ca^2+^-free CaM (apo-CaM) causes the unmasking of hydrophobic pockets in the lobes. Ca^2+^-CaM (holo-CaM) binds to receptor sequences containing positively charged amphiphilic alpha-helices, interacting with them electrostatically and hydrophobically.

### 4.1. Calmodulin Binding Domains in Connexins

In 1981 Hertzberg and Gilula first reported that CaM binds to Cx32 [65]. Soon after, their data were confirmed for gap junctions of bovine eye lens and crayfish hepatopancreas [64]. CaM binding to Cx32 was also supported by data showing that CaM protects Cx32 from m-calpain-induced proteolysis [88] and prevents phosphorylation of Cx32 by the tyrosine kinase of the EGF receptor [89]. CaM interaction with connexins was well supported by in vitro experimental data demonstrating that CaM participates in the oligomerization of Cx32 monomers in connexons [90].

Several years ago, we identified two CaM sites in Cx32: one at the NH2 domain (NT) and one at the initial COOH domain (CT1) [91]. Six years later, Török et al. reported that CaM interacts with peptides containing the CaM-binding sequences of Cx32’s NT and CT1, by testing a fluorescent derivative of CaM (TA-CaM) [92] and equilibrium fluorescence techniques [82]. The peptides interacted with TA-calmodulin in a Ca^2+^-dependent manner [82]. Later on, lobe-specific binding of CaM to peptides of Cx32 was tested by stopped-flow kinetics, employing CaM mutants that are Ca^2+^-binding-deficient [81]. Peptides corresponding to the NT domain of Cx32 (res. 1–22) interacted with NH2- and COOH-terminal lobes of CaM (N- and C-lobes). The C-lobe interacted with higher affinity, while peptides matching the CT1 sequence (res. 208–227) bound to either CaM lobe one at a time [81].

We studied the predicted CaM-binding sites of the NT and CT1 sequences of 13 connexins by a computer program (http://calcium.uhnres.utoronto.ca/ctdb/ctdb/sequence.html). Several connexins (Cx26, 31, 31.1, 32, 33, 40, and 43) were found to have a CaM-binding site at NT (Figure 6) and only a few (Cx31, 32, 36, and 43) at CT1 (Figure 7).

CaM binding to Cx32’s CT1 was confirmed by isothermal titration calorimetry (ITC) and nuclear magnetic resonance (NMR) [93]. More recently, CaM was also found to bind to Cx43’s CT domain (res. 264–290) [94]. The CT domains of Cx35 and Cx36 are relevant to channel gating [95], while those of Cx43 and Cx32 are unlikely to be involved, as Cx43’s CT deletion at res. 257 [96] and Cx32’s CT deletion by 84% [97] or 100% [98] do not alter the sensitivity of chemical gating.

CaM was reported to interact with the CT1 domain of mouse Cx35, Cx36, and Cx34.7 [99,100]. CaM-Cx36 interaction at CT1 was further confirmed by NMR, which demonstrated that CaM interacts with CT1 in a conventional compact conformation to a mostly hydrophobic eight-residue sequence (res. 277–284) [100]. The Cx36-CaM complex formed before the assembly of Cx36 in gap junctions and allowed for cell–cell diffusion of a dye [100]. CaM blockers or mutation of the residue W277, which is important for the interaction of CaM and Cx36, blocked cell–cell diffusion of a dye [100]. The interaction of CaM with Cx36 preceded gap junction formation, confirming evidence for the role of CaM in connexin assembly into gap junctions [90]. The importance of CT1 for Cx35 channel-gating was confirmed by Aseervatham and coworkers [95].

In 1996, in oocyte pairs of Xenopus, we started testing the gating sensitivity to CO_2_ of channels composed of chimeras and mutants of Cx32 and Cx38, two types of channel that are opposite in chemical gating sensitivity [101,102]. In fact, 3 min application of 100% CO_2_ decreases Gj of Cx38 channels to zero, whereas it decreases Gj of Cx32 channels by only ~15% (Figure 8B).

Channels made of a Cx32/38 CL chimera with the Cx32’s CL domain replaced with that of Cx38 (Figure 8A) precisely reproduced the chemical gating sensitivity of Cx38 channels in uncoupling magnitude and speed of uncoupling and recoupling (Figure 8B). In contrast, Cx32/38NT channels (Cx32’s NT replaced with that of Cx38) behaved closer to Cx32 channels [102]. These data demonstrated for the first time the relevance of the CL domain in determining chemical gating sensitivity [102].

In order to more precisely identify the CL sequence most important for chemical gating sensitivity, we expressed Cx32/Cx38 chimeras with either the first half (CL1) or the second half (CL2) of Cx38’s CL replacing those of Cx32 [101] (Figure 8A). The CO2 sensitivity of channels made of Cx32/Cx38CL2 (Cx32 with Cx38’s CL2) matched that of Cx38 channels in CO2 sensitivity, but Gj recovered faster than with the channel made of Cx38 (Figure 8B). Cx32/Cx38CL1 (Cx32 with Cx38’s CL1) could not be tested because functional channels did not express. The data suggest that CL2 contains a sequence most relevant to chemical gating [101]. CL2’s role in sensitivity of chemical gating is consistent with evidence from studies that have identified a CL2 CaM-binding site in Cx43, Cx50, and Cx44 (rev. in [12]) in addition to Cx32, Cx35, Cx45, and Cx57 [83,103]. CL2 is probably the most relevant CaM-binding site because CL2’s CaM-binding sites are predicted to be present in all of the 13 murine connexins tested (Figure 9). However, it should be stressed here that the analysis of the predicted CaM-binding sites of the NT, CT1, and CL2 sequences by a computer program (http://calcium.uhnres.utoronto.ca/ctdb/ctdb/sequence.html) involves only linear amino acid sequences. Indeed, future work is needed to further test the CaM–connexin interaction by the AlphaFold analysis [104], as this would go beyond simple predictions of the binding sites as presented in Figure 6, Figure 7 and Figure 9. Furthermore, attempts should be made to perform AlphaFold3 [105] analysis as this would even allow the inclusion of Ca^2+^ in the prediction.

Our data on the role of CL2 domain in chemical gating [101] were first confirmed by evidence for CaM binding to CL2 of Cx43 (res. 136–158) [85], as a peptide containing this domain was found to bind to Ca^2+^-CaM with a stoichiometry of 1:1, when tested by surface plasmon resonance, circular dichroism, fluorescence spectroscopy, and NMR [85]. Far-UV circular dichroism experiments demonstrated that upon CaM binding, the in α-helical content of the peptide increases [85]; this was further proven in fluorescence and NMR experiments, which showed that the conformation of both peptide and CaM changes as they form the CaM–peptide complex. The apparent dissociation constant (KD) of peptide–CaM interaction in physiologic [K+]i ranged from 0.7 to 1 µM. As the peptide bound to CaM, the KD of Ca^2+^ for CaM decreased from 2.9 ± 0.1 µM to 1.6 ± 0.1 µM, and the Hill coefficient (nH) increased from 2.1 ± 0.1 to 3.3 ± 0.5 [85].

For testing gating efficiency of channels made of mutants of Cx43 that lack the CaM-binding domain, two mutants linked to the fluorescent protein EYFP were tested in HeLa cells. The absence of CL2’s CaM-binding site abolished the Ca^2+^-dependent gating sensitivity, confirming that CL2 (res. 136–158) possesses the CaM-binding site important for Ca^2+^-dependent gating [85].

CL2’s relevance as a CaM-binding site was further proven by experiments testing channels made of Cx43 [38], Cx50 [86], or Cx44 [84], rev. in [12]. A study [106] used a synthetic peptide corresponding to Cx43’s CL2’s CaM-binding domain (res. 144–158) to test for conformational changes in the Ca^2+^-CaM–peptide complex by small angle X-ray scattering. With peptide binding, the dumbbell shape of CaM was lost, and CaM became more globular, indicating that CaM binds to the peptide in a conventional “collapsed” structure [106].

Xu et al. [38] studied N2a cells that expressed human Cx43 or Cx40 by whole-cell patch clamp. With ionomycin addition, [Ca^2+^]_i_ increased threefold and Gj decreased by 95%; the Ca^2+^-induced drop in Gj was prevented by application of the CaM inhibitor CDZ and was reversed by adding 10 mM EGTA to a Ca^2+^-free solution [38]. Addition of a Cx43 peptide corresponding to the CL2’s CaM-binding domain (res. 136–158) to the pipette solutions abolished gating as well, while a scrambled peptide or the Ca^2+^/CaM-dependent kinase II’s inhibitory peptide (res. 290–309) did not [38]. The data prove that the CaM-binding domain of CL2’s is a major player in Cx43’s gating. A summary of the predicted CaM-binding sites of thirteen murine connexins is shown in Figure 10.

### 4.2. CaM Is Linked to Connexins at Resting [Ca^2+^]i

Many studies have reported that CaM is linked to connexins at resting [Ca^2+^]_i_. Recently, this has been supported by studies that tested CaM binding to peptides corresponding to the CL2 CaM-binding domain of Cx32, Cx35, Cx45, and Cx57, in the presence or absence of Ca^2+^ [83]. Fluorescence changes in the FRET-probe DA-CaM and the Ca^2+^-sensitive TA-CaM were monitored by fluorescence spectroscopy and stopped-flow fluorimetry [92] at physiological ionic strength (pH 7.5, 20 °C). FRET measurements showed partial compaction of DA-CaM (54–70% quenching in the presence of Ca^2+^ and 33–62% quenching in its absence). The kinetic data revealed a two-step sequence: fast interaction and isomerization. These data confirmed that CaM is linked to connexins and binds fully to them with Ca^2+^ activation [83]. Significantly, the CL2 peptides of Cx45 and Cx57 bind to CaM with high affinity both with and without Ca^2+^, suggesting that CaM is linked by the C-lobe either in Ca^2+^-free or in Ca^2+^-bound state.

### 4.3. CaM–Connexin Co-Localization

CaM–connexin binding was also studied by immune fluorescence microscopy [78,79]. In HeLa cells expressing Cx32, CaM, and Cx32 co-localized at cell–cell contact (Figure 11) [78,79]. Cells expressing Cx43 or Cx37 gave similar data (Sotkis and Peracchia, unpublished), and so did cells expressing Cx50 [107,108] or Cx36 [100].

Recently, the interaction between CaM and Cx45 was visualized in vivo by BRET (bioluminescence resonance energy transfer) [80]; the interaction of CaM and Cx45 required Ca^2+^ and was prevented by W7, a CaM inhibitor. CaM-Cx45 interaction involved the CL2 CaM-binding site (residues 164–186; Figure 9). This was further proven by data demonstrating the high-affinity interaction of CaM fluorescently labeled with a peptide corresponding to the CL2 domain [80]. Indeed, we have provided data for both Ca^2+^-dependent and -independent CaM interaction with Cx45’s CL2 sequence [83]. The Ca^2+^-independent CaM-CL2 interaction confirms evidence that CaM is linked to connexins at normal [Ca^2+^]_i_ (~50 nM) [78,79,80,109].

## 5. Cork-Gating Model

Twenty-four years ago, we suggested a “cork-type” CaM-mediated gating model [78,110]. The model involves two mechanisms [111]. One, “Ca-CaM-Cork”, proposes a Ca^2+^-induced blockage of the connexon’s pore by CaM’s N-lobe (also likely to involve connexin conformational changes), following the interaction of the N-lobe to a connexin’s site. The other, “CaM-Cork”, also suggests a blockage of the channel’s pore by CaM’s N-lobe, but without activation by Ca^2+^ [112]. In the first, channel gating is reversed by the recovery of [Ca^2+^]_i_ to resting values. In the second, gating is reversed by positive Vj at the gated side.

The Ca-CaM-cork gating model is based on data accumulated over the last quarter century, which include the following:Behavior of connexin mutants [76,78,113];Behavior of single-channel gating [61];Behavior of channels expressed in oocytes over-expressing CaMCC, a mutant with higher Ca^2+^ sensitivity [78,79];Vj sensitivity of the chemical gate [114];Evidence that each negatively charged CaM lobe is ~25 × 35 Å in diameter (Figure 12) [87], which is the same as the size of the positively charged intracellular channel mouth (Figure 12) [115,116,117].

Among others, rev. in [19,111].

This Ca-CaM-Cork model suggests that with a rise in [Ca^2+^]_i_ above resting values, the CaM’s N-lobe is activated and this enables it to interact with a CaM site and plug the channel’s mouth (Figure 12) [19,109,111,112]. At resting [Ca^2+^]_i_, CaM is believed to be bound to each connexin of a connexon by the C-lobe, most likely at the CL2 site [83]. In contrast, the N-lobe is believed to be free. The Ca^2+^-binding constant of the C-lobe of CaM is greater than that of the N-lobe by nearly one order of magnitude [118,119]. Thus, our model proposed that the N-lobe binds to the connexin’s gating site (CL2 or NT) when [Ca^2+^]_i_ increase above basal values. This model is consistent with evidence for an independent function of the two lobes in binding to Cx32 [81].

It should be kept in mind, however, that even at resting state, some gap junction channels are closed [120]. They may be closed by the CaM-cork mechanism or, perhaps, even by the Ca-CaM-cork mechanism. Indeed, evidence that addition of 20 mM BAPTA to patch pipette solutions greatly enhances cell–cell coupling in astrocytes [33] suggests that some channels may even be sensitive to resting [Ca^2+^]_i_ [33].

With a [Ca^2+^]_i_ rise, a possible scenario is that the N-lobe of each of the six CaM anchored to the connexon is activated, interacts with either the NT or the CL2 site of the same connexin and changes the conformation of the connexon, enabling one of the six N-lobes to enter into the channel’s mouth and plug the pore by interacting with the NT or CL2 site of another connexin. A different scenario may be that while all N-lobes are activated, one of them wins the competition, interacts with NT or CL2 of the opposite connexin, and plugs the pore by binding hydrophobically and electrostatically to the site (Figure 12).

## 6. Conclusions and Future Perspectives

Data accumulated in the past four decades have strongly supported the role of Ca^2+^-CaM in gap junction channel gating. Indeed, as detailed in previous chapters, evidence that CaM mediates the Ca^2+^_i_ role in gating is based on data generated by multiple studies, which have demonstrated the effectiveness of CaM blockers, CaM expression/inhibition, expression of a CaM mutant with greater Ca^2+^-sensitivity, co-localization of CaM and connexins at gap junctions, evidence that connexins contain high-affinity CaM-binding sites, behavior of connexin mutants, evidence that gating is activated by the application of large and repeated Vj pulses negative at the gating side, gating behavior of single channels, recovery of gating efficiency by addition of Ca^2+^-CaM to intracellularly perfused crayfish axons, X-ray diffraction findings in isolated gap junction fragments, among others; rev. in [12,13,19,121].

Despite all these data on Ca-CaM gating, a recent publication has challenged the Ca-CaM mechanism by proposing a direct “electrostatic” Ca^2+^ role in gap junction channel gating [10,11]. While this model may by relevant for the gating mechanism of connexin hemichannels, as they are exposed to [Ca^2+^]o in the millimolar range, [Ca^2+^]_i_ as high as 20 mM, as tested in this study [11], by far exceed the well-known effectiveness of high nanomolar to low micromolar [Ca^2+^]_i_ in channel gating; rev. in [13,19]. Furthermore, since the “no-Ca-added saline” used for control condition did not contain Ca^2+^-chelators, the [Ca^2+^] of this control saline must have been greater than 12 µM, which is significantly higher that the high nM to low µM [Ca^2+^]_i_ sufficient to activate channel gating (see in the previous). Indeed, evidence that both pore diameter and channel conformation are unaltered even with application of such unphysiologically high [Ca^2+^]_i_ [10,11] further strengthens the requirement of a Ca^2+^-modulated protein (CaM) for activating Ca^2+^-gating of gap junction channels.

It is very important to understand in detail the gating mechanism of gap junction channels because cell uncoupling is not only a safety mechanism for preventing healthy cell to be damaged by broken neighboring cells (healing-over), but also because a Ca^2+^-gating sensitivity in the high nM range suggests that the fine regulation of cell coupling is likely to be relevant for maintaining tissue homeostasis. In the future, researchers in the gap junction field should also consider testing the effect on gap junction function of CaM mutants. Indeed, recent evidence that certain CaM mutants cause a variety of diseases [122,123,124,125,126,127] points to a potential role of CaM mutants in causing diseases that result from abnormal cell communication via gap junction channels.

Thus far, CaM mutations have been reported to cause cardiac diseases; most mutations are found in CaM’s C-lobe, one in CaM’s N-lobe and one in the C-lobe-to-N-lobe linker. CaM mutations most frequently result in long QT syndrome (LQTS) [128], a change that alters cardiac electrical activity, often resulting in the catecholaminergic polymorphic ventricular tachycardia (CPVT) phenotype [129] and idiopathic ventricular fibrillation (IVF). Patients with CPVT display ventricular tachycardia, frequently resulting in ventricular fibrillation and death. Most often, cardiac diseases result from the action of CaM mutants on the ryanodine receptor (RyR2) and the L-type voltage-gated Ca^2+^ channel, but other channels are likely to be involved as well. Therefore, it is important to consider testing CaM mutants on gap junction mediated cell–cell communication as well.

Knowledge of the pivotal role of Ca-CaM in modulating the function of connexin channels may also pave the way for understanding the behavior of hemichannels and intracellular connexin/innexin channels. Indeed, evidence indicates that connexins form hexameric connexons in non-junctional plasma membranes, the Golgi apparatus, mitochondria, and the endoplasmic reticulum; rev. in [130]. Many questions need to be answered, such as: Are connexons capable forming functional intracellular hemichannels? Do they interact to form “intracellular” junctions? Do they interact with gap junctions? Indeed, interesting findings published in the last four decades have suggested that connexin-mediated communication might also take place intracellularly between organelles, as well as between organelles and gap junctions. Furthermore, the role of Ca-CaM in diseases caused by connexin and/or CaM mutations (rev. in [131]) needs to be addressed in detail.

## Figures and Tables

**Figure 1 ijms-25-09789-f001:**
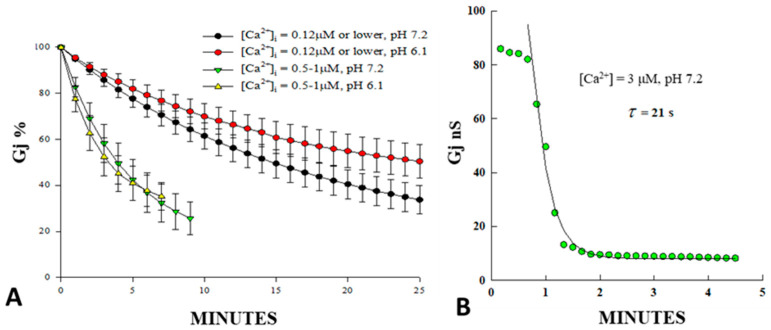
Gj of Novikoff hepatoma cell pairs internally dialyzed via patch pipettes filled with solutions well buffered for H^+^ and Ca^2+^. [Ca^2+^]_i_ = 0.12 μM or lower, caused Gj to drop to 40-50% of initial values with τ = 35.2 and 22.3 min, at pHi = 6.1 and 7.2, respectively (**A**). This is the normal Gj decay in cells studied by whole-cell patch clamp. [Ca^2+^]_i_ = 0.5–1.0 μM, caused Gj to decrease to 25% of initial values with τ’s = 5.9 and 6.2 min, at pHi = 6.1 and 7.2, respectively (**A**). [Ca^2+^]_i_ = 3 μM (pH = 7.2) uncoupled the cells in less than 1 min with τ = ~21 s (**B**). This confirms that the gating mechanism is insensitive to a cytosolic acidification of pH 6.2, if [Ca^2+^]_i_ is carefully buffered with BAPTA. Adapted from Ref. [30].

**Figure 2 ijms-25-09789-f002:**
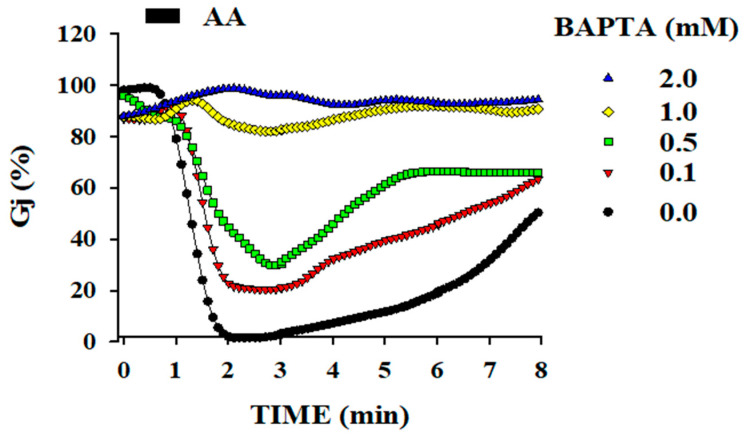
Gj of Novikoff hepatoma cell pairs exposed for 20 s to 20 mM arachidonic acid (AA) while being internally dialyzed via patch pipettes containing solutions buffered for Ca^2+^ with BAPTA (pH = 7.2). The drop of Gj is completely prevented by buffering [Ca^2+^]_i_ with BAPTA. Note that even BAPTA concentrations as low as 0.1 mM are effective. Adapted from Ref. [1].

**Figure 3 ijms-25-09789-f003:**
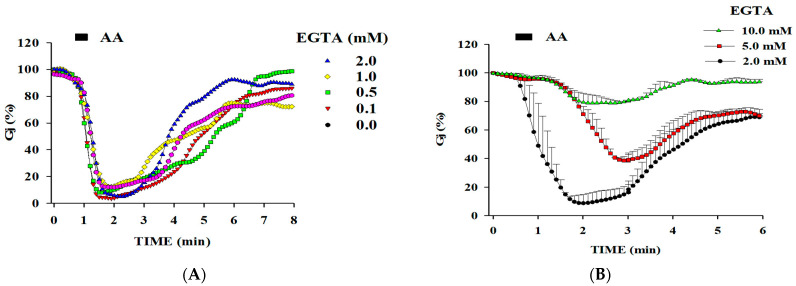
Gj of Novikoff hepatoma cell pairs exposed for 20 s to 20 mM arachidonic acid (AA) while being internally dialyzed via patch pipettes containing solution buffered for Ca^2+^ with different [EGTA] ((**A**,**B**); pH = 7.2). EGTA is 10 times less effective than BAPTA (see Figure 2) in inhibiting the AA effect on Gj. This is consistent with evidence that EGTA is significantly less efficient than BAPTA in buffering [Ca^2+^]_i_. Adapted from Ref. [1].

**Figure 4 ijms-25-09789-f004:**
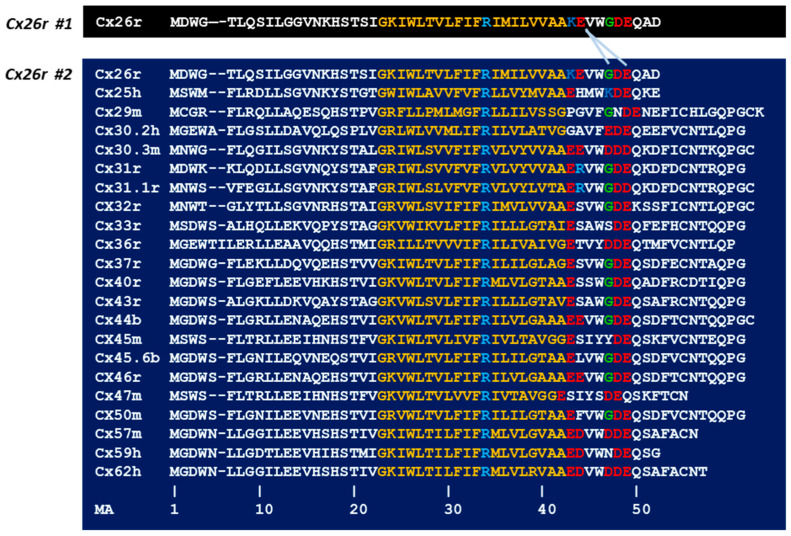
Multiple amino acid alignment (MA) of mammalian connexins in the domain spanning from the NH2 terminus to the initial sequence of the first extracellular loop (EL1). The electrostatic Ca^2+^ gating model proposes that Ca^2+^ links adjacent connexin monomers at 3 Ca^2+^ sites located at the NH2 terminus end of the E1. These sites involve two residues of one Cx26 monomer (G45 and E47) and one residue (E42) of the adjacent monomer (see arrows). The Cx labels “r, h, m, and b” are acronyms of “rat, human, mouse, and bovine”, respectively.

**Figure 5 ijms-25-09789-f005:**
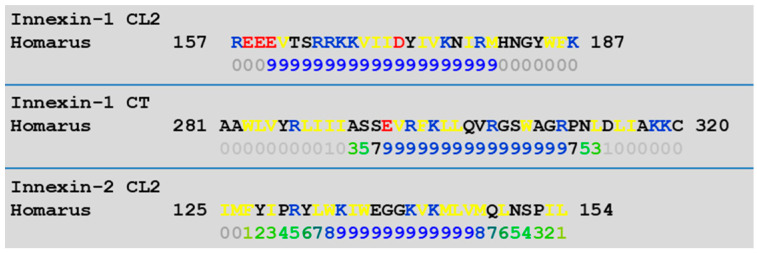
CaM-binding predictions at CT and CL2 domains of innexins-1 and -2 (in blue letters), identified by a computer program (http://calcium.uhnres.utoronto.ca/ctdb/ctdb/sequence.html, accessed on 29 January 2013).

**Figure 6 ijms-25-09789-f006:**
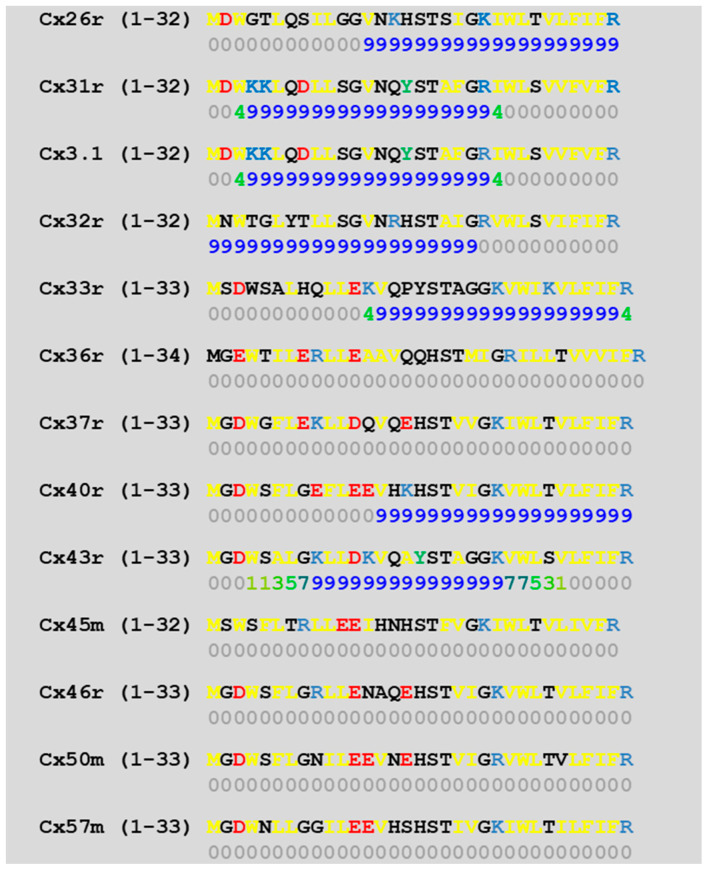
Predicted CaM-binding sites at connexins’ NH2 terminus domain (NT) (in blue letters), identified by a computer program (http://calcium.uhnres.utoronto.ca/ctdb/ctdb/sequence.html).

**Figure 7 ijms-25-09789-f007:**
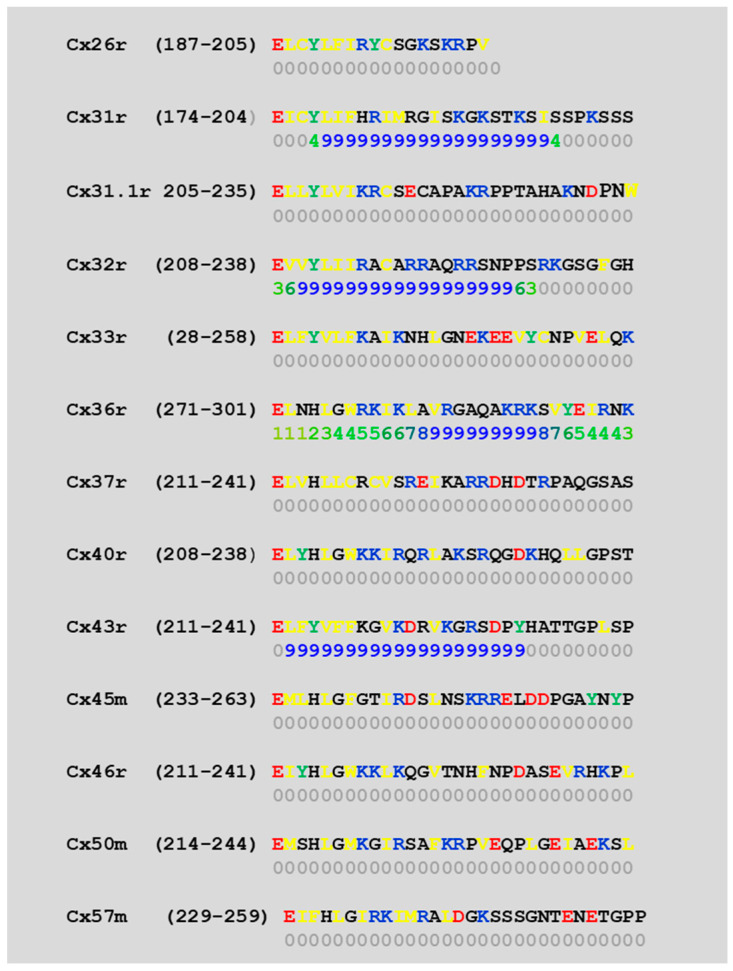
Predicted CaM-binding sites at connexins’ initial COOH terminus domain (CT1) (in blue letters), identified by a computer program (http://calcium.uhnres.utoronto.ca/ctdb/ctdb/sequence.html). Note that only 4 of these 13 connexins have a potential CaM-binding site.

**Figure 8 ijms-25-09789-f008:**
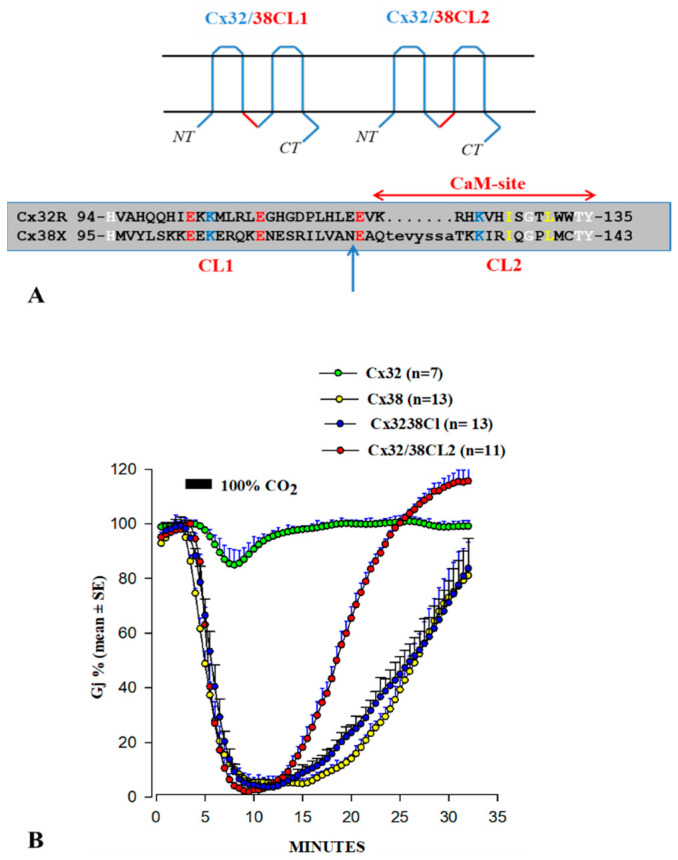
Gj drop caused by superfusion of saline gassed with 100% CO_2_ in pairs of *Xenopus* oocytes expressing Cx32, Cx38, or Cx32/38 chimeras. Cx32/38CL channels (Cx32’s CL replaced with that of Cx38 (**A**)) or Cx32/38CL2 (Cx32’s CL2 replaced with that of Cx38 (**A**)) match the gating sensitivity of Cx38 channels (**B**), but Gj recovers faster with Cx32/38CL2 (**B**). Adapted from Ref. [101].

**Figure 9 ijms-25-09789-f009:**
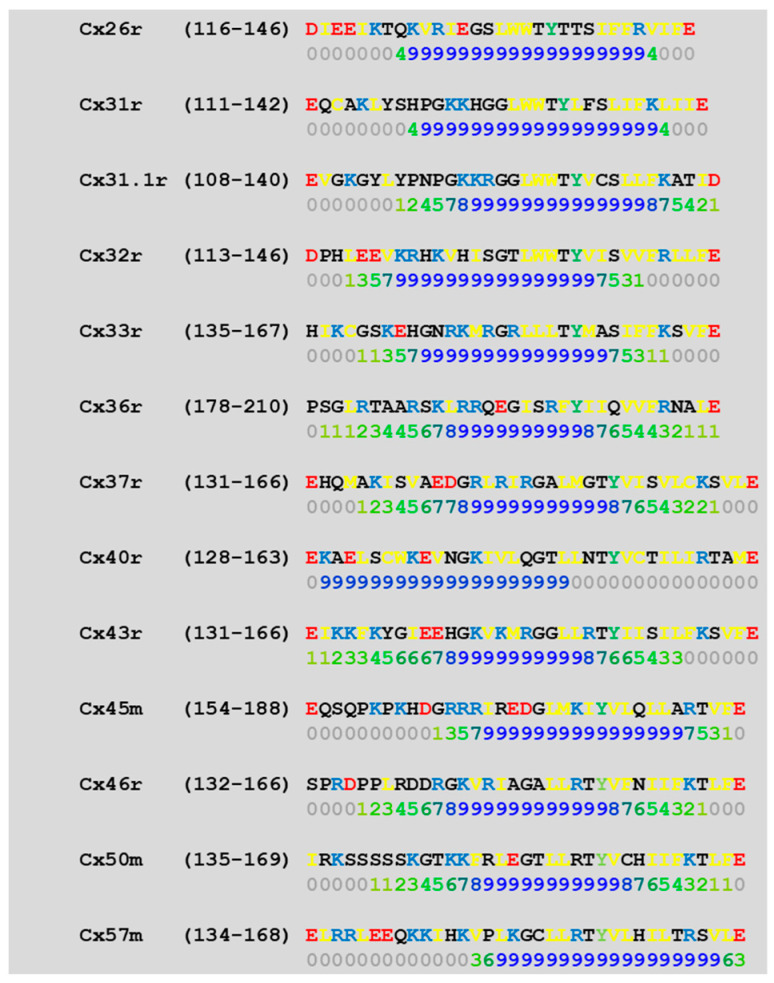
Predicted CaM-binding sequences of the second half of the cytoplasmic loop (CL2) of connexins (in blue letters), identified by a computer program (http://calcium.uhnres.utoronto.ca/ctdb/ctdb/sequence.html). Note that all these 13 connexins have a potential CaM-binding site.

**Figure 10 ijms-25-09789-f010:**
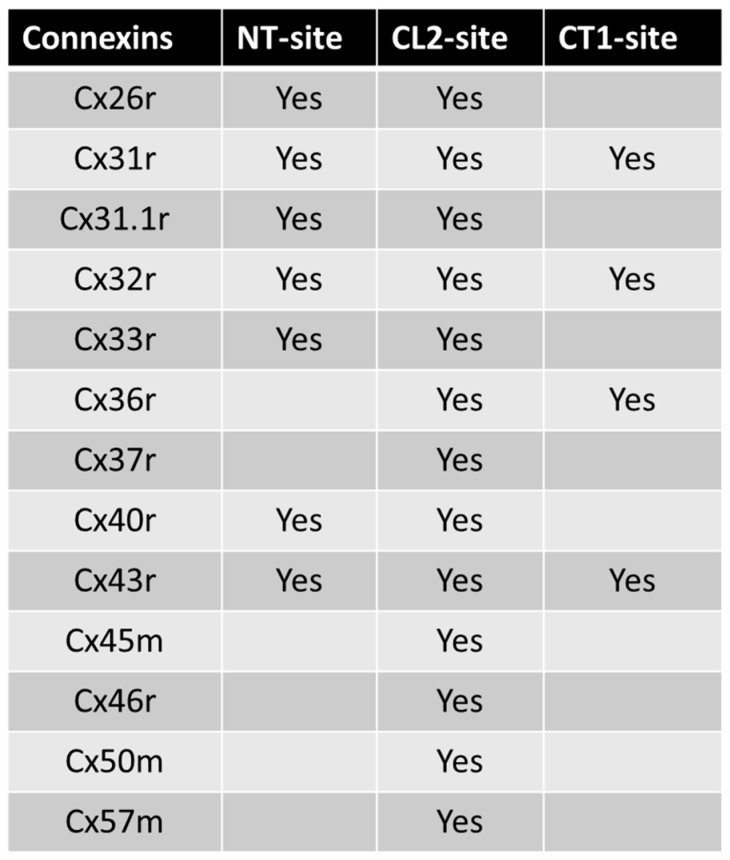
Predicted CaM-binding sites at connexins’ NT, CL2, and CT1 domains.

**Figure 11 ijms-25-09789-f011:**
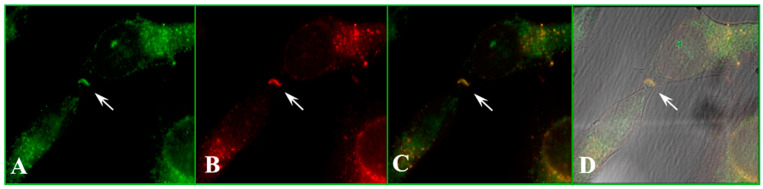
Immunofluorescence microscopy of HeLa cells stably transfected with Cx32 and sequentially labeled for Cx32 and CaM. (**A**,**B**) show labeling for CaM and Cx32, respectively. (**C**) shows the overlay of (**A**,**B**), and (**D**) adds to the overlay the bright field image. Note the colocalization of CaM and Cx32 at the at the junctional site (arrow) and at most, but not all, of the cytoplasmic spots.

**Figure 12 ijms-25-09789-f012:**
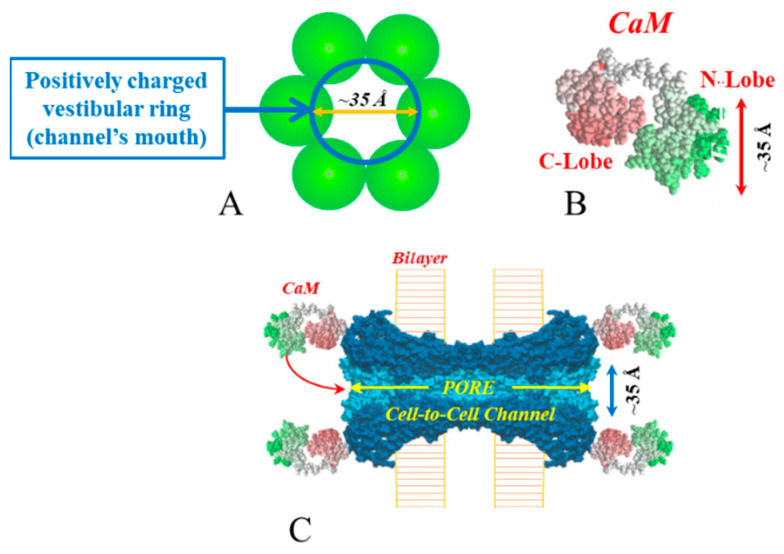
The positively charged channel’s vestibule (**A**,**C**) and the negatively charged lobes of CaM (**B**) are ~25 × 35 Å in diameter. Thus, a CaM lobe could fit well within the positively charged connexon’s vestibule (**A**,**C**). In **C**, the channel is cut along its length to show the pore’s diameter (light blue area) throughout its entire length. CaM and connexon images (**B**,**C**) were generously provided to us by Drs. Francesco Zonta and Mario Bortolozzi (VIMM, University of Padua, Italy).

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
