# Peer review of "Calcium Role in Gap Junction Channel Gating: Direct Electrostatic or Calmodulin-Mediated?"

_ijms, 2024, doi:10.3390/ijms25189789_

Round 1

Reviewer 1 Report

Comments and Suggestions for Authors

The manuscript by Peracchia, “Calcium role in gap junction gating. Direct electrostatic or calmodulin-mediated?” reviews the literature and the alternative hypothesis on the mechanism of calcium effects on connexin channel conformations and gating. This is an authoritative review of the literature by an expert in this field of research. I only have minor comments that hopefully would help the author in improving this review.

The most substantial request / suggestion is as follows: the author should consider including, in addition to his bioinformatics illustrations featuring the CaM binding sites, the AlphaFold predictions of CaM-Cx peptide interactions. Theoretically this could be quite informative, and would go beyond mere predictions of the binding sites as presented currently in the figures - i.e. if there is indeed a good chance the the NT, CL or CT domains of Cx proteins can be engaged by CaM, AlphaFold mind just be able to predict that. One could attempt to do AlphaFold predictions with peptides slightly larger than the target sequences shown in the figures, together with CaM. AlphaFold3 would even allow the user to include Ca2+ ions in the prediction. Whether a prediction of this type could be corroborated is another question - but this could be an interesting point of this review, potentially enticing the scientists in this field to look into this experimentally.

Apart from this I have a few small comments on the wording:

Abstract: “changes are present” - not clear what this means. Presumably the author meant the changes are not present, but one could read that the pore-diameter is not present (which does not make sense).

Figure 5. “..identified by a program”

The final paragraph might benefit from a few references (i.e. referring to CaM mutations, long QT syndrome, CPVT, IVF, Ryr2, etc.).

Throughout the text the author writes “in previous” - it might be good to clarify what previous is referred to in each case (previous sentence, paragraph, paper, etc).

Comments on the Quality of English Language

The English is fine.

Author Response

I appreciate the detailed and thoughtful comments of the reviewer, and I have made all efforts to address them.

The most substantial request / suggestion is as follows: the author should consider including, in addition to his bioinformatics illustrations featuring the CaM binding sites, the AlphaFold predictions of CaM-Cx peptide interactions. Theoretically this could be quite informative, and would go beyond mere predictions of the binding sites as presented currently in the figures - i.e. if there is indeed a good chance the the NT, CL or CT domains of Cx proteins can be engaged by CaM, AlphaFold mind just be able to predict that. One could attempt to do AlphaFold predictions with peptides slightly larger than the target sequences shown in the figures, together with CaM. AlphaFold3 would even allow the user to include Ca2+ ions in the prediction. Whether a prediction of this type could be corroborated is another question - but this could be an interesting point of this review, potentially enticing the scientists in this field to look into this experimentally.

We agree with the reviewer and have added the following sentence: “However, it should be stressed here that the analysis of the predicted CaM-binding sites of the NT, CT1 and CL2 sequences by a computer program (http://calcium.uhnres.utoronto.ca/ctdb/ctdb/sequence.html) involves only linear amino acid sequences. Indeed, future work is needed to further test the CaM-connexin interaction by the AlphaFold analysis (Jumper, et al., 2021), as this would go beyond simple predictions of the binding sites as presented in figures 6, 7 and 9. Furthermore, attempts should be made to perform AlphaFold3 analysis (Abramson et al., 2024) as this would even allow the inclusion of Ca2+ in the prediction”.

While it might not be a good excuse, the reviewer should realize that the author has only minimal experience in structural biology, has retired almost 2 decades ago and no longer has research funds.

Apart from this I have a few small comments on the wording:

Abstract: “changes are present” - not clear what this means. Presumably the author meant the changes are not present, but one could read that the pore-diameter is not present (which does not make sense).

The last sentence has been modified as follows: “However, this study, which tested the effect of unphysiologically high [Ca2+]i on the structure of isolated junctions, reported that neither changes in the channel’s pore-diameter nor connexin conformational changes are present, in spite of exposure of isolated gap junctions to [Ca2+]i as high at the 20 mM”.

Figure 5. “..identified by a program” Done

The final paragraph might benefit from a few references (i.e. referring to CaM mutations, long QT syndrome, CPVT, IVF, Ryr2, etc.).

We have added two reference (Prakash et al 2022, 2023) to the total of 8 references to original papers and review articles. (see below).

  1. Nyegaard, M.; Overgaard, M. T.; Sondergaard, M. T.; Vranas, M.; Behr, E. R.; Hildebrandt, L. L.; Lund, J.; Hedley, P. L.; Camm, A. J.; Wettrell, G.; Fosdal, I.; Christiansen, M.; Borglum, A. D., Mutations in calmodulin cause ventricular tachycardia and sudden cardiac death. Am J Hum Genet 2012, 91, (4), 703-12.
  2. Jensen, H. H.; Brohus, M.; Nyegaard, M.; Overgaard, M. T., Human Calmodulin Mutations. Front Mol Neurosci 2018, 11, 396.
  3. Badone, B.; Ronchi, C.; Kotta, M. C.; Sala, L.; Ghidoni, A.; Crotti, L.; Zaza, A., Calmodulinopathy: Functional Effects of CALM Mutations and Their Relationship With Clinical Phenotypes. Front Cardiovasc Med 2018, 5, 176.
  4. Kotta, M. C.; Sala, L.; Ghidoni, A.; Badone, B.; Ronchi, C.; Parati, G.; Zaza, A.; Crotti, L., Calmodulinopathy: A Novel, Life-Threatening Clinical Entity Affecting the Young. Front Cardiovasc Med 2018, 5, 175.
  5. Chazin, W. J.; Johnson, C. N., Calmodulin Mutations Associated with Heart Arrhythmia: A Status Report. Int J Mol Sci 2020, 21, (4).
  6. Crotti, L.; Spazzolini, C.; Tester, D. J.; Ghidoni, A.; Baruteau, A. E.; Beckmann, B. M.; Behr, E. R.; Bennett, J. S.; Bezzina, C. R.; Bhuiyan, Z. A.; Celiker, A.; Cerrone, M.; Dagradi, F.; De Ferrari, G. M.; Etheridge, S. P.; Fatah, M.; Garcia-Pavia, P.; Al-Ghamdi, S.; Hamilton, R. M.; Al-Hassnan, Z. N.; Horie, M.; Jimenez-Jaimez, J.; Kanter, R. J.; Kaski, J. P.; Kotta, M. C.; Lahrouchi, N.; Makita, N.; Norrish, G.; Odland, H. H.; Ohno, S.; Papagiannis, J.; Parati, G.; Sekarski, N.; Tveten, K.; Vatta, M.; Webster, G.; Wilde, A. A. M.; Wojciak, J.; George, A. L.; Ackerman, M. J.; Schwartz, P. J., Calmodulin mutations and life-threatening cardiac arrhythmias: insights from the International Calmodulinopathy Registry. Eur Heart J 2019, 40, (35), 2964-2975.
  7. Prakash, O.; Gupta, N.; Milburn, A.; McCormick, L.; Deugi, V.; Fisch, P.; Wyles, J.; Thomas, N. L.; Antonyuk, S.; Dart, C.; Helassa, N., Calmodulin variant E140G associated with long QT syndrome impairs CaMKIIdelta autophosphorylation and L-type calcium channel inactivation. J Biol Chem 2023, 299, (1), 102777.
  8. Prakash, O.; Held, M.; McCormick, L. F.; Gupta, N.; Lian, L. Y.; Antonyuk, S.; Haynes, L. P.; Thomas, N. L.; Helassa, N., CPVT-associated calmodulin variants N53I and A102V dysregulate Ca2+ signalling via different mechanisms. J Cell Sci 2022, 135, (2).

Throughout the text the author writes “in previous” - it might be good to clarify what previous is referred to in each case (previous sentence, paragraph, paper, etc).

Changed to “in the previous chapter” or “in a previous chapter”

Reviewer 2 Report

Comments and Suggestions for Authors

In this manuscript entitled “Calcium role in gap junction channel gating. Direct-electrostatic or calmodulin-mediated?”, the author reviewed literature about chemical gating of gap junction channels mediated by cytosolic Ca2+i and about the role of Calmodulin, a Ca2+-binding protein. The author presents an extensive and detailed review of the literature, integrating historical studies with recent findings, offering a well-rounded understanding of the topic. The inclusion of figures showing molecular interactions, such as the mechanism by which CaM binds to connexins, helps to visualize the complex mechanisms discussed. Of particular interest is the detailed exploration of CaM binding sites within connexins and the analysis of their significant role in the assembly of connexins into functional gap junctions with a particular focus on the light chain 2 (CL2) domain of connexins. The crucial involvement of CaM in the gating mechanism is therefore emphasized.

The author demonstrates, through his own research as well as that of others, that the CL2 domain plays a critical role in determining the chemical gating sensitivity of connexins, specifically Cx43. The review is very interesting and relevant for this area of interest.

Here follows minor comments that should be addressed:

-              In the abstract, the sentence ‘’Indeed, the role of Ca2+-CaM in gating is well supported by studies that have tested CaM-blockers, CaM expression inhibition, testing of CaM mutant with higher Ca2+ sensitivity, co-localization of CaM and connexins, existence of CaM-binding sites with high affinity in connexins/innexins, expression of mutants of connexins (Cx) with greater gating sensitivity, whose gating efficiency is eliminated by inhibition of CaM expression, among others.’’ is too long and intricate to understand. I suggest simplifying this sentence to improve the readability.

-              The abstract lacks a concluding sentence and a final statement. I suggest closing the abstract with a conclusive take-home message. 

-              In page 7, I propose to remove the sentence from the article [14], since it does not provide any additional important information, and the inclusion of original references from article [14] might be confusing.

-              While the article focuses on elucidating the specific mechanism of Ca2+-CaM in gap junction gating, a brief discussion in the conclusion section, of the broader physiological and pathophysiological implications of these findings, especially in relation to disease state, would be beneficial to provide a context for the relevance of the topic in the research field, beyond the molecular aspect 

-              Abbreviations should be revised, since some terms should be introduced more clearly at first mention.

Comments on the Quality of English Language

In my opinion the quality of English is good and does not require any particular revisions although I have the appropriate qualification to assess it.

Author Response

I appreciate the detailed and thoughtful comments of the reviewer, and I have made all efforts to address them.

In this manuscript entitled “Calcium role in gap junction channel gating. Direct-electrostatic or calmodulin-mediated?”, the author reviewed literature about chemical gating of gap junction channels mediated by cytosolic Ca2+i and about the role of Calmodulin, a Ca2+-binding protein. The author presents an extensive and detailed review of the literature, integrating historical studies with recent findings, offering a well-rounded understanding of the topic. The inclusion of figures showing molecular interactions, such as the mechanism by which CaM binds to connexins, helps to visualize the complex mechanisms discussed. Of particular interest is the detailed exploration of CaM binding sites within connexins and the analysis of their significant role in the assembly of connexins into functional gap junctions with a particular focus on the light chain 2 (CL2) domain of connexins. The crucial involvement of CaM in the gating mechanism is therefore emphasized.

The author demonstrates, through his own research as well as that of others, that the CL2 domain plays a critical role in determining the chemical gating sensitivity of connexins, specifically Cx43. The review is very interesting and relevant for this area of interest.

Here follows minor comments that should be addressed:

-              In the abstract, the sentence ‘’Indeed, the role of Ca2+-CaM in gating is well supported by studies that have tested CaM-blockers, CaM expression inhibition, testing of CaM mutant with higher Ca2+ sensitivity, co-localization of CaM and connexins, existence of CaM-binding sites with high affinity in connexins/innexins, expression of mutants of connexins (Cx) with greater gating sensitivity, whose gating efficiency is eliminated by inhibition of CaM expression, among others.’’ is too long and intricate to understand. I suggest simplifying this sentence to improve the readability.

The sentence has been shortened as follows: “Indeed, the role of Ca2+-CaM in gating is well supported by studies that have tested CaM-blockers, CaM expression inhibition, testing of CaM mutants, co-localization of CaM and connexins, existence of CaM-binding sites in connexins/innexins, expression of connexins (Cx) mutants, among others.”

-              The abstract lacks a concluding sentence and a final statement. I suggest closing the abstract with a conclusive take-home message.

Thi sentenced has been added: “In conclusion, data generated in the past four decades by multiple experimental approaches have clearly demonstrated the direct role of Ca2+-CaM in gap junction channel gating.” 

-              In page 7, I propose to remove the sentence from the article [14], since it does not provide any additional important information, and the inclusion of original references from article [14] might be confusing.

Done

-              While the article focuses on elucidating the specific mechanism of Ca2+-CaM in gap junction gating, a brief discussion in the conclusion section, of the broader physiological and pathophysiological implications of these findings, especially in relation to disease state, would be beneficial to provide a context for the relevance of the topic in the research field, beyond the molecular aspect

The following paragraph has been added at the end, although I am not sure that it fully addresses the reviewer comment. Sorry!

Knowledge of the pivotal role of Ca-CaM in modulating the function of connexin channels may also pave the way for understanding the behavior of connexin hemichannels, intracellular connexin/innexin channels. Indeed, evidence indicates that connexins form hexameric connexons in non-junctional plasma membranes, the Golgi apparatus, mitochondria and the endoplasmic reticulum; rev. in [131]. Many questions need to be answered, such as: are connexons capable forming functional intracellular hemichannels? Do they interact to form “intracellular” junctions? Do they interact with gap junctions? Indeed, interesting findings published in the last four decades, have suggested that connexin-mediated communication might also take place intracellularly between organelles, as well as between organelles and gap junctions. Furthermore, the role of Ca-CaM in diseases caused by connexin and/or CaM mutations (rev. in [132]) needs to be addressed in detail.

       Abbreviations should be revised, since some terms should be introduced more clearly at first mention.

Done.